# Adverse Event Burden Score—A Versatile Summary Measure for Cancer Clinical Trials

**DOI:** 10.3390/cancers12113251

**Published:** 2020-11-04

**Authors:** Jennifer G. Le-Rademacher, Shauna Hillman, Elizabeth Storrick, Michelle R. Mahoney, Peter F. Thall, Aminah Jatoi, Sumithra J. Mandrekar

**Affiliations:** 1Department of Health Sciences Research, Mayo Clinic, Rochester, MN 55905, USA; hillman.shauna@mayo.edu (S.H.); storrick.elizabeth@mayo.edu (E.S.); mandrekar.sumithra@mayo.edu (S.J.M.); 2IQVIA, Raleigh-Durham, NC 27703, USA; mahoney.michelle@mayo.edu; 3Department of Biostatistics, M.D. Anderson Cancer Center, Houston, TX 77230, USA; rex@mdanderson.org; 4Department of Oncology, Mayo Clinic, Rochester, MN 55905, USA; jatoi.aminah@mayo.edu

**Keywords:** adverse event, clinical trials, adverse event burden score, safety profile

## Abstract

**Simple Summary:**

In cancer clinical trials, adverse event data are collected after every treatment cycle, using the Common Terminology Criteria for adverse events, which includes 837 terms. The vast number of potentially reportable adverse events over multiple treatment cycles makes summarizing and analyzing adverse event data challenging. The current standard reporting of adverse event data includes the frequency of the maximum (worst) grade of commonly occurring adverse events. In this article, we propose a single quantitative summary measure that incorporates both the frequency and the severity of multiple adverse events over time; the adverse event burden score. This score is a well-defined measure that enables statistical comparisons analogous to other quantitative endpoints in clinical trials. The adverse event burden score can readily accommodate different trial settings, diseases, and treatments, with diverse safety profiles.

**Abstract:**

This article introduces the adverse event (AE) burden score. The AE burden by treatment cycle is a weighted sum of all grades and AEs that the patient experienced in a cycle. The overall AE burden score is the total AE burden the patient experienced across all treatment cycles. AE data from two completed Alliance multi-center randomized double-blind placebo-controlled trials, with different AE profiles (NCCTG 97-24-51: 176 patients, and A091105: 83 patients), were utilized for illustration. Results of the AE burden score analyses corroborated the trials’ primary results. In 97-24-51, the overall AE burden for patients on the treatment arm was 2.2 points higher than those on the placebo arm, with a higher AE burden for patients who went off treatment early due to AE. Similarly, in A091105, the overall AE burden was 1.6 points higher on the treatment arm. On the placebo arms, the AE burden in 97-24-51 remained constant over time; and increased in later cycles in A091105, likely attributable to the increase in disease morbidity. The AE burden score enables statistical comparisons analogous to other quantitative endpoints in clinical trials, and can readily accommodate different trial settings, diseases, and treatments, with diverse AE profiles.

## 1. Introduction

The collection of adverse events (AEs) is an important aspect of cancer clinical trials, with the goal of capturing “untoward medical events” that cancer patients experience while enrolled on trials. The reporting of such AEs is often mandated by the National Cancer Institute (NCI) and the US Food and Drug Administration. Guidance for the reporting of AEs is detailed in the NCI Guidelines for Investigators [1]. Classification and grading of AEs are reported using the Common Terminology Criteria for Adverse Events (CTCAE), currently using version 5.0 [2]. The CTCAE includes 837 AE terms for 26 system/organ classes, with most AEs classified into five severity grades (1 = mild, 2 = moderate, 3 = severe, 4 = life threatening, and 5 = death). Adverse events are typically assessed and reported immediately after each cycle of treatment.

The vast number of potentially reportable AEs, coupled with an often large number of treatment cycles, make summarizing and analyzing AE data challenging. Consequently, AEs are often summarized in a descriptive manner. Typically, the frequency of the maximum (worst) grade of the most common AEs reported during the course of treatment are tabulated, or compared between treatment arms in a randomized trial using frequency of grade 3 or worse AEs while ignoring the types of events, or sometimes classifying the AEs as hematologic or non-hematologic. 

It is useful to have a single measure that reflects the overall AE burden by including all AEs graded and reported during a trial, and that can be applied across trials to facilitate AE profile comparisons, and/or serve as benchmark for new trials. In this paper, we introduce a unified framework for defining such a measure, the AE burden score. This statistic is similar to the total toxicity burden (TTB), introduced by Bekele and Thall [3] in the context of dose-finding in a phase I trial in sarcomas [4], and the longitudinal version of TTB defined by Hobbs et al [5] in the context of designing a randomized trial of chemoradiation +/− surgery for esophageal cancer, and is also similar to the severity score used by Schuurhuizen et al [6].

The definition of AE burden score given here is flexible, to facilitate its use in qualitatively different settings. Since the types and the grades of AEs, as well as their impacts on patients, can vary widely across tumor types, treatments, and trial settings, in practice the proposed summary measure should be defined a priori to be clinically meaningful and ensure its objectivity. To illustrate application of this measure, we use data from two completed multi-center cancer clinical trials, with distinct AE profiles computed for each treatment arm in each trial.

## 2. Methods

The primary goal of the methodology that follows is to describe the single quantitative variable that summarizes the frequencies and severities of multiple AEs that may occur over time in cancer patients undergoing therapy. Below, we provide a general definition of AE burden score. A list of all symbols used as mathematical notation in this section is included at the end of the paper for reference.

### 2.1. Defining AE Burden Measures and Associated Analysis Methods

Let $Y_{kg}$(*t*) be the indicator that a patient experiences an AE of type k (= 1, …, *K*), grade g(=1,…,5), where g=5 indicates death due to AE type k, at time (or treatment cycle) t. That is,
(1)$$Y_{kg}\left(t\right)=\{\begin{array}{cc} 1, & \mathrm{if}\;\mathrm{the}\;\mathrm{patient}\;\mathrm{experiences}\;\mathrm{event}\;k\;\mathrm{of}\;\mathrm{grade}\;g\;\mathrm{at}\;\mathrm{time}\;t\\ 0, & \mathrm{otherwise}. \end{array}$$

Using these indicator variables, multiple measures of AE burden can be defined. Specific examples, including the current convention of occurrence of grade 3 or worse AEs, are given in Appendix A. All of these approaches reduce the multi-dimensional AE information into a categorical summary measure.

We now introduce the framework for computing an AE burden at each time point by computing a weighted sum of $Y_{kg}\left(t\right)$ values over all grades and AEs of interest. The definition requires a pre-specified severity weight, $w_{kg}$, for each combination of adverse event k and grade g. The values of $(w_{kg}$, *k* = 1, …, *K*, g = 1, …, 5) are subjective, and they serve the dual purposes of quantifying how bad one considers a combination (*k*, *g*) to be, and putting the severities of qualitatively different toxicities on the same numerical domain. For example, all values of $w_{kg}$ may be specified on a domain of 0 to 10 or 0 to 100 for convenience. In practice, the severity weights may be elicited from oncologists familiar with the treatments and AEs for a particular type of cancer being studied, and if desired, consensus values may be obtained from a group of stakeholders.

We define the patient AE burden at time t as
(2)$$B\left(t\right)=\sum_{k}\sum_{g}w_{kg}Y_{kg}\left(t\right).\;$$

The AE burden B(t) is thus a quantitative variable. This is similar to expressions (1) and (2) of Hobbs et al [5] for quantifying possibly recurrent toxicities. The burden at each time point t can be summarized using mean (standard deviation), and/or median (range); compared between the treatment arms using a *t*-test or the Mann–Whitney test; expressed as a function of treatment/dose and patient covariates using a regression model for each *t*, or more generally used as repeated measures across all time points, t, and analyzed by a longitudinal regression model.

An overall AE burden score across treatment cycles/times can be defined by summing over *t* as:(3)$$TB=\sum_{t}u_{t}\left[\sum_{k}\sum_{g}w_{kg}Y_{kg}\left(t\right)\right].\;$$

Note that *TB* is a single quantitative summary reflecting the overall AE burden that a patient experiences across all treatment cycles, henceforth referred as the ***overall AE burden score***, in contrast with *B(t)* which is the AE burden at a single treatment cycle *t*. The overall AE burden, *TB*, can be compared between treatment arms and modeled using parametric or non-parametric methods for continuous outcomes.


*Some considerations of the AE burden score:*
*Weight functions*: The weight functions $u_{t}$ (weight for time t) and $w_{kg}$ (weight for AE type and grade) should be defined a priori in a manner that is relevant to the disease, treatments, and study objectives. Interpretation of the AE burden at time, *t*, B(t), and the overall AE burden score, TB, depend on these weight functions. Specifically:
-A simple weight function for $w_{kg}$ can be $w_{kg}=g$, i.e., the weight of an AE equals the grade of the event regardless of *k*. In this case, the interpretation of B(t) is the total of all the grades across all adverse events that a patient experienced at time t. Although a limitation of this definition is that, for example, *g* = 3 for two qualitatively different AEs *k* and *r* have the same weight $w_{k3}=w_{r3}=$ 3, it is easily interpretable and informative in most settings. This weight function also takes advantage of the work that has already gone into the development of the CTCAE, where the AE grades reflect similar severity from one AE type to another. A more complex weight function, if needed, may be defined a priori with consensus from stakeholders, including clinicians, patients, and others. -For $u_{t}$, a *TB* with the weight function $u_{t}=1$ equates to the total grades of AEs across all treatment cycles that a patient experienced. A *TB* with the weight function $u_{t}=1/c$, where *c* is the number of treatment cycles the patient received, equates to the overall AE burden a patient experienced averaged across all treatment cycles.*Grade 5 events:* With grade 5 indicating the worst outcome of death from the AE, one may argue that the impact of a grade 5 event is much more burdensome relative to that of lower grade events. Therefore, the weight for grade 5 events can be inflated relative to the weights of grade 1-4 events as deemed appropriate for a specific tumor type and/or trial setting. However, when comparing AE burdens across trials, it is important that the same weight function be used to ensure comparability across trials. In our analyses of the two completed trials presented in the Results section, below, we assigned a weight of 10 to grade 5 events, using the weight domain from 0 to 10. This choice of weight for grade 5 events was intended to reflect the increased burden of death due to the AE (counts as twice its severity), while at the same time not being too large that it overshadows the burden of lower grade events. *Missing data assumptions:* All solicited AEs are evaluated at every treatment cycle; for all other AEs, we assume that they are not present if not recorded for that cycle.


### 2.2. The Utility of AE Burden Scores

Table 1 shows the proposed burden scores alongside the maximum grade (current standard reporting of AEs) for 9 patients who had different AE profiles. Although Patients 1–3 experienced only mild AEs (grade 1), their AE burdens, *B(t)*, vary slightly by treatment cycle, and the total overall burdens (*TB*) for these patients range from 2 to 7 (average overall burden 1 to 1.4). Using the current reporting standard, all 3 patients would have the same maximum AE grade of 1. More noticeably, Patients 4–6 had some moderate AEs (grade 2) with varying *B(t)* from cycle to cycle, and the total overall burden, *TB*, for these patients range from 12 to 59 (average overall burden 2.4 to 19.7). Using the current reporting standard, these 3 patients would have the same maximum AE grade of 2. Thus, the burden scores incorporate information on the number and the severity of the AEs that a patient reported and provide a much more informative summary of the patient’s experience than the simple measure of maximum grade during treatment.

Note that in Table 1, the overall AE burden averaged across cycles as well as the total overall AE burden are presented. Both measures provide a summary of the overall AE burden. In studies where most patients are expected to stay on treatment for a similar duration, and early dropout is not a major concern, analyses using either the average or the total score should yield similar conclusions. However, in situations where patients are expected to go off treatment at different time points, the choice of whether to choose the average or the total overall burden requires careful consideration, as they have different implications. The total overall AE burden does not adjust for the number of treatment cycles a patient received, or the important possibility that treatment was discontinued due to AEs. If the number of treatment cycles is predefined and it is likely that all patients will have the same number of treatment cycles, then the total overall AE burden might be an appropriate measure to use. In situations where patients are expected to go off treatment early if a severe AE occurs, or treatment is given until disease progression, it is important to account for the number of treatment cycles a patient received. In this case, the average overall AE burden would be a more appropriate measure to use. For example, consider Patients 7–9 (Table 1). Although the total overall AE burden of Patient 7 (27) is only slightly higher than those of Patients 8 and 9 (26 for both), the average overall AE burden of Patient 7, who went off treatment at cycle 2, was 13.5 compared to 6.5 and 5.2 for Patients 8 and 9, who went off treatment at cycles 4 and 5, respectively. Although the total overall AE burden scores were comparable, Patient 7 experienced a much higher average overall AE burden than Patients 8 and 9.

### 2.3. Application to Clinical Trials

The proposed AE measures were applied to data from two completed clinical trials, NCCTG 97-24-51 [7] and A091105 [8], conducted through the Alliance for Clinical Trials in Oncology, an NCI-funded national clinical trials network. It is important to emphasize that the AE burden analyses of these trials and the results presented here are intended only for illustration of the proposed measure. These results are not intended to replace the results reported in the primary trial publications. The analyses conducted for this manuscript used de-identified data, and do not require ethical approval nor informed consent.

The trials included in this paper were chosen for two reasons. First, both trials were randomized double-blind, and placebo-controlled, and thus the placebo arms serve as references to evaluate toxicity of the active treatments by comparing the AE burden of the treatment arms to that of the placebo arms. The placebo arms also serve as potential benchmarks when evaluating the AE burden scores for future trials in the same disease and setting. Thus, our first hypothesis was that the AE burden score would be higher in the active treatment arm compared to the placebo arm in each trial. Second, we wanted to illustrate application of the AE burden to trials in different diseases. This diversity of disease types and disease behavior (one an aggressive malignancy and the other a non-malignant, slow growing cancer) allowed us to assess AE burden when the diseases themselves were generating AEs. Thus, our second hypothesis was that the AE burden score would capture the disease-induced AEs, both in the case of patients with aggressive malignancy (non-small cell lung cancer) as well as in patients who had less-aggressive desmoid tumors.

## 3. Results

### 3.1. Overview of the Two Selected Trials

The two trials selected for illustrating this AE measure were NCCTG 97-24-51 and A091105. The first trial included 176 patients, with 85 randomized to active agent as part of maintenance therapy with carboxyaminotriazole (CAI), which was proven to be therapeutically ineffective, and 91 to placebo [7]. The second trial included 83 patients, with 47 randomized to sorafenib, which significantly prolonged progression-free survival, and 36 to placebo [8].

The AEs in both trials were aligned with expectations based on the administered agents/placebos and the patients’ underlying diseases. In NCCTG 97-24-51, non-hematologic AEs were mostly grade 1 or 2, with consistently higher rates in the CAI arm, including fatigue (54.5% versus 29.3%), anorexia (31.1% versus 13.0%), nausea (62.2% versus 30.4%), vomiting (32.2% versus 14.1%), neurosensory (60.0% versus 44.6%), and ataxia (33.3% versus 16.3%) [7]. In A091105, higher rates of grade 3–4 AEs were reported in the sorafenib arm (47%) compared to the placebo arm (25%). A much greater percentage of patients on the sorafenib arm experienced palmar-plantar erythrodysesthesia syndrome (71% versus 22%) and rash/skin disorder (87% versus 42%) compared to the placebo arm [8].

### 3.2. Understanding the Overall AE Burden Score, TB, in the Context of Placebo Arms

The AE burden by treatment cycle, *B(t)*, with weight, $w_{kg}$, equaling the AE grade for grade 1–4 events, and a weight of 10 for grade 5 events, as well as the overall burden score across treatment cycles (*TB* with weight $u_{t}=1/c$) were computed for each patient, as described earlier. To benchmark AE burden scores based on the placebo arm, we assessed scores within each trial by treatment arm. For NCCTG 97-24-51, the overall AE burden score for patients who received CAI was higher than those who received placebo (median *TB*: 5 versus 2.8, Wilcoxon *p*-value <0.0001; Table 2). Of note, patients on the CAI arm went off treatment more quickly than patients on the placebo arm (the number of patients on treatment at each cycle is shown at the bottom of Figure 1A), especially those with high AE burden in the early cycles (as shown in Figure 1B). Nonetheless, the overall AE burden score for patients on the CAI arm was 2.2 points higher than for patients on the placebo arm. Specifically, Figure 1A shows that the AE burden by cycle in the CAI arm was high in the early period (cycle 1 mean = 6.2, 95% confidence interval (CI): 5.2–7.2, n = 85) and decreased over time (cycle 2 mean = 4.8, 95% CI: 3.6–6.1, n = 52; cycle 5 mean = 1.7, 95% CI: 0.84–2.5, n = 16), whereas the AE burden by cycle of patients on the placebo arm remained relatively constant over time (cycle 1 mean = 3.7, 95% CI: 2.7–4.7, n = 91; cycle 2 mean = 3.8, 95% CI: 2.9–4.7, n = 75; cycle 5 mean = 3.6, 95% CI: 2.0–5.1, n = 28).

Similarly, for A091105, the AE burden at each cycle, *B(t)*, and the overall AE burden score, *TB*, using the same weight functions as described for NCCTG 97-24-51, were computed for each patient. As the majority of patients stayed on treatment for a long time (48% received 15 cycles of treatment), we did not evaluate the pattern of AE burden scores by duration of treatment. Figure 2 shows that the AE burden by cycle was consistently higher in the sorafenib arm compared to the placebo arm; the overall AE burden score across treatment cycles was 1.6 points higher with sorafenib compared to placebo (median *TB*: 3.6 versus 2.0, Wilcoxon p-value = 0.0042; Table 2). The AE burden by cycle of patients on the placebo arm was relatively flat in the early treatment cycles, but showed an increase in later cycles (Figure 2), with these observations likely attributable to the disease morbidity experienced by patients on the placebo arm. A higher proportion of patients on the sorafenib arm remained on treatment by cycle 15 compared to the placebo arm (53% versus 42%) despite the higher overall AE burden score on the sorafenib arm; this pattern appears to reflect improved disease control.

### 3.3. Clinically Logical Patterns of the AE Burden by Treatment Cycle, B(t)

Given the high rate of patients going off treatment in NCCTG 97-24-51, we sought to understand the patterns of association between the AE burden by cycle and patients’ duration on treatment (as shown in Figure 1B). Only 16 patients remained on the CAI arm by cycle 5. Figure 1B shows a higher AE burden by cycle experienced by patients who went off treatment early, as expected. The AE burden by cycle for the 49 patients who received only one cycle of CAI treatment was higher than those who went off treatment in subsequent cycle (mean *B(t)*: 7.2 versus 2.6 to 5.2). These findings of high AE burden followed by a rapid withdrawal from the trial are in keeping with what one would expect to see in clinical practice, and they illustrate how the AE burden may be used to quantify this pattern.

Furthermore, Figure 3 shows the AE burden by cycle grouped by treatment arm, and whether patients went off treatment due to AE or by choice versus due to disease progression or other reasons. The small number of patients who went off treatment for other reasons were combined with disease progression. The AE burden by cycle for patients who went off treatment due to AE or by choice in the CAI arm (mean *B(t)* between 8 and 9) was much higher than the AE burden of patients who went off treatment due to disease progression or other reasons (mean *B(t)* 2 to 5 for the CAI arm, and mean *B(t)* 3 to 4 for the placebo arm).

Similar patterns were observed in A091105. Figure 4 shows the AE burden by cycle, grouped by the reason patients went off treatment by arm. In the sorafenib arm, patients who went off treatment due to AE or by choice had higher AE burden compared to those who remained on treatment. The AE burden of these patients on the sorafenib arm was higher than the AE burden of patients on the placebo arm, who went off treatment due to disease progression or other reason reasons. Again, these analyses illustrate how the AE burden by cycle captures patterns of study withdrawal within the context of desmoid tumors, a non-malignant, less-aggressive disease entity.

## 4. Discussion

This paper describes a framework to define an AE burden measure that is simple, yet flexible enough to accommodate different trial settings, diseases, and treatments; accommodating diverse adverse event safety profiles. It uses all of the information collected on AEs during the trial, unlike other descriptive summaries that provide only maximum (worst) grade over time, and that include only adverse events relevant to specific categories of AE. In essence, the overall AE burden score provides a comprehensive picture of the AE burden experienced by patients, appears to be more informative than more commonly used approaches, and yet manages to distill complex data into a single score.

Of note, other tools have been devised to summarize adverse event data. For example, the Toxicity over Time (ToxT) [9,10] is such tool, which uses a combination of statistical techniques, ranging from graphical summary to repeated measures models to survival analysis, to summarize the adverse event profile over the entire course of study. The ToxT provides a major improvement over more conventional adverse event reporting. However, in contrast to the AE burden score, the ToxT requires more extensive synthesis of data and more extensive explanation of the methodology that led to the stated conclusions. Such long explanations preclude succinct reporting of clinical trial data, which typically need to also focus on multiple endpoints, not toxicity alone. Again, an advantage of the AE burden score is its succinctness.

A major advantage of the proposed AE burden score appears to be its functionality within a specific clinical trial. First, because it is quantitative, it allows for comparisons between trial arms, thus allowing investigators to make inferences about safety and tolerability of an investigational intervention based on a single well-defined statistic, as opposed to multiple verbal descriptors of adverse events by category. The use of a single overall AE burden score, as opposed to sorting through several verbal categories of adverse events, facilitates formal comparison between study arms. Second, the AE burden score readily captures AE severity. The score quantifies the magnitude of the difference in AE burden between the arms of the trial, making it an ideal measure of the severity of treatment toxicity. Third, the AE burden score is well defined and easy to understand. It incorporates all reported AE data and distills a sizable amount of data into a single score. Thus, this aspect of the reporting of AE differences between arms can be reduced to a statistical comparison in a manner analogous to comparisons of survival or other similar endpoints in clinical trials [11,12,13]. 

In addition to within-trial comparisons, the AE burden score could also facilitate comparisons of AEs across trials for the same disease and patient population, as long as the same weighing schemes are used for AEs. Furthermore, a baseline AE burden can be established as an anchor, for example, for defining a threshold for acceptable toxicity for various diseases or for various patient populations. With the establishment of such anchors, the use of AE burden score can be applied to single arm trials where the AE burden of an investigational agent can be compared against the established anchor for that disease setting and patient population, again, using the same weighing schemes. This approach might also lead to greater safety in monitoring clinical trials in a real time manner, as adverse event data that approach an *a priori* established anchor threshold could prompt a more timely review of a trial for safety purposes.

As with any method, appropriate use of AE burden score requires care and consideration. First, although the proposed measure is flexible, it should be clearly defined a priori to ensure its objectivity and its comparability across trials. Since the purpose of this manuscript is to introduce the AE burden score and to illustrate its application, we did not compare the use of different weight functions. The weight function used in our examples is simple and readily interpretable, and it may be applied quite generally in many different settings. However, if a more complex weight function is preferable, the weight functions should be considered carefully and agreed upon by all stakeholders, to provide a clinically meaningful interpretation. Of note, the same weight function should be used if AE burden scores are compared across trials. Second, it is important to note that AE burden score cannot ameliorate the problem of poor quality data. More specifically, with more than 800 AE terms and multiple treatment cycles, not all AEs are reported at every cancer treatment cycle. In well-designed trials, AEs that are clinically relevant are required to be evaluated and reported, commonly called solicited AEs or AEs of special interest. However, missing data or the sporadic capture of AE data can lead to compromised study conclusions, as is the case with any endpoint that relies on only partially collected data. Thus, it remains prudent to closely evaluate the patterns and mechanism of missing data to ensure that appropriate statistical methods are used when analyzing AE burden scores. This is a complex issue that may be explored in future applications.

## 5. Conclusions

With all of the foregoing considerations in mind, the proposed AE burden score provides a simple and objective approach to the current reporting and analysis of AE data. It quantifies the magnitude of the AE burden that patients experience during their cancer treatment and should be considered a safety endpoint in cancer clinical trials. This measurement merits further research to determine optimal weight functions, ongoing testing, and further integration into cancer clinical trials over time.

## Figures and Tables

**Figure 1 cancers-12-03251-f001:**
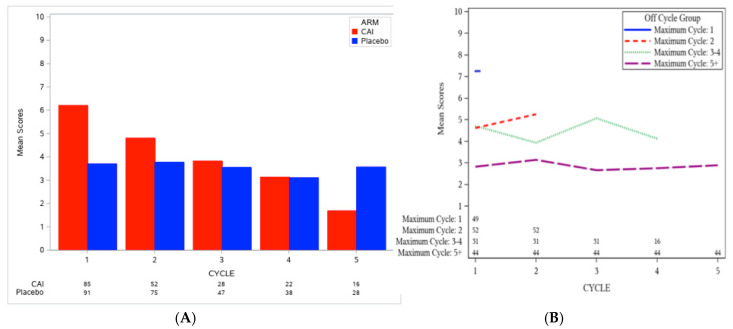
(**A**) AE burden by treatment cycle, *B(t)*, grouped by treatment arm; (**B**) AE burden by treatment cycle, *B(t)*, (across treatment arms) grouped by the last administered cycle for NCCTG 97-24-51.

**Figure 2 cancers-12-03251-f002:**
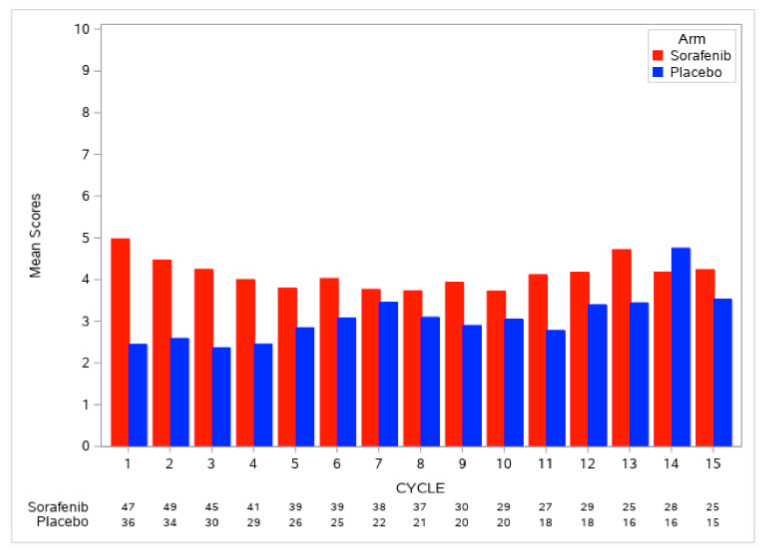
AE burden by treatment cycle, *B(t)*, grouped by treatment arm for A091105.

**Figure 3 cancers-12-03251-f003:**
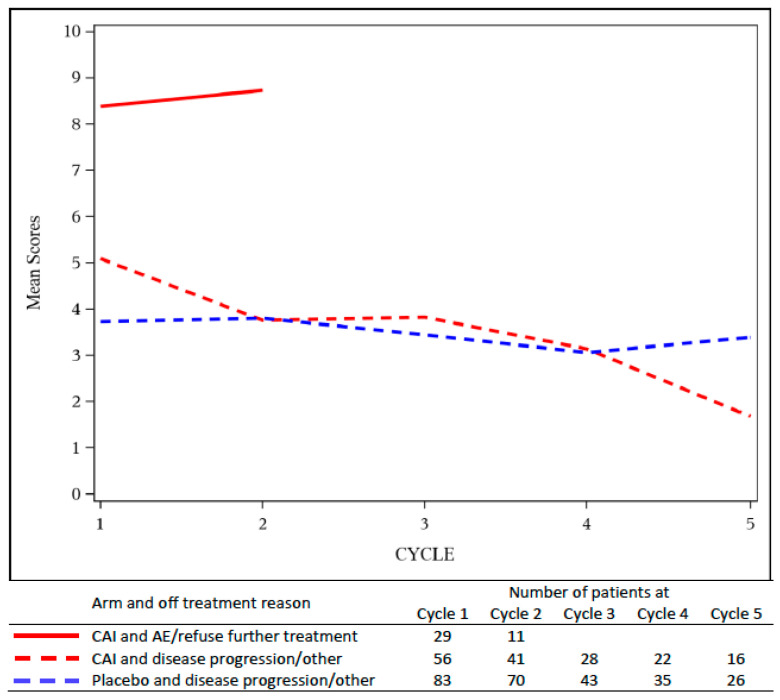
AE burden by treatment cycle, *B(t)*, grouped by treatment arm and by reason patient went off treatment for NCCTG 97-24-51. The AE burden of CAI patients who went off treatment due to adverse events (solid line) was higher than that of patients on either arm who went off for disease progression (dashed lines).

**Figure 4 cancers-12-03251-f004:**
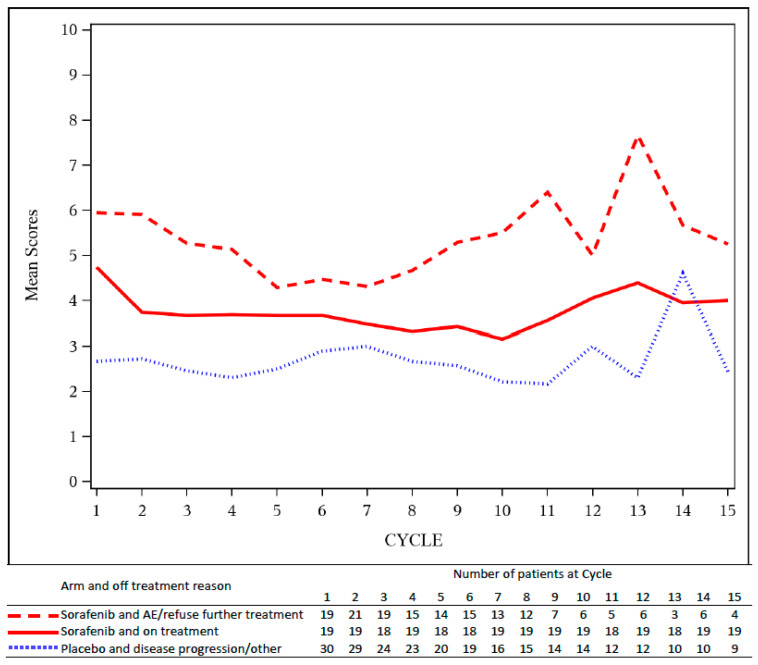
AE burden by treatment cycle, *B(t)*, grouped by off-treatment reason and treatment arm for A091105. The AE burden of patients on sorafenib who went off treatment due to adverse events (dashed red line) was higher than the AE burden of patients on sorafenib who remained on treatment (solid red line) or those on placebo who went off for disease progression (dotted blue line).

**Table 1 cancers-12-03251-t001:** Adverse event (AE) burden scores for various AE profiles (from real patients) compared to the current reporting standard of maximum grade (Mild: Grade 1, Moderate: Grade 2, Severe: Grade 3+, early termination: cycle 1–2, mid termination: cycle 3–4, late termination: cycle 5–6).

Patient Scenario	AE Profile	AE Burden Score (new)							Maximum Grade (Current Standard)
		*B(t)*–by Cycle					*TB*–Overall Across All Cycles		
		1	2	3	4	5	Average	Total	
**1. Mild** AEs with early treatment termination	Cycle 1: grd 1 alopecia. Cycle 2: grd 1 creatinine. Off treatment cycle 2.	1	1	NA	NA	NA	1	2	1
**2. Mild** AEs with mid treatment termination	Cycle 1: grd 1 alopecia. Cycle 2: grd 1 alopecia. Cycle 4: grd 1 alopecia and anorexia. Off treatment cycle 4.	1	1	0	2	NA	1	4	1
**3. Mild** AEs with late treatment termination	Cycle: grd 1 neuro-sensory. Cycle 2: grd 1 neuro-sensory and alopecia. Cycle 3: grd 1 neuro-sensory and alopecia. Cycle 4: grd 1 neuro-sensory. Cycle 5: grd 1 neuro-sensory. Off treatment cycle 5.	1	2	2	1	1	1.4	7	1
**4. Moderate** AEs with early treatment termination	Cycle 1: grd 1 alopecia, alkaline phosphatase, anorexia, cough, edema, and anemia; grd 2 headache. Cycle 2: grd 1 alopecia, alkaline phosphatase, SGPT (ALT), and anemia; grd 2 headache, ataxia, and pain-bone. Off treatment cycle 2	8	10	NA	NA	NA	9	18	2
**5. Moderate** AEs with mid treatment termination	Cycle 1: grd 1 alopecia, constipation, anxiety, insomnia, neuro-motor, depression, taste, pain-abdominal, urinary frequency, auditory, and neuro-sensory; grd 2 dyspnea, stomatitis, arthralgia, and myalgia. Cycle 2: grd 1 dysphagia, anxiety, depression, fatigue, pain-abdominal, headache, insomnia, arthralgia, neuro-sensory, auditory, neuro-motor, constipation, and renal; grd 2 dyspnea. Cycle 3: grd 1 anorexia, voice change, injection site reaction, anxiety, renal, taste, depression, dysphagia, cough, myalgia, and stomatitis; grd 2 dyspnea, neuro-motor, neuro-sensory, vomiting, nausea, fatigue, and pain-abdominal. Off treatment cycle 3.	19	15	25	NA	NA	19.7	59	2
**6. Moderate** AEs with late treatment termination	Cycle 1: grd 1 cough; grd 2 rhinitis allergic, and pain-bone. Cycle 3: grd 1 alopecia and cough. Cycle 4: grd 1 pain; grd 2 nausea and infection without neutropenia. Off treatment cycle 5.	5	0	2	5	0	2.4	12	2
**7. Severe** AEs with early treatment termination	Cycle 1: grd 1 neuro-sensory and weight loss; grd 2 arthralgia and fatigue. Cycle 2: grd 1 anemia; grd 2 anorexia and cough; grd 3 dyspnea and pain-chest; grd 5 adult respiratory distress syndrome. Off treatment cycle 2.	6	21	NA	NA	NA	13.5	27	5
**8. Severe** AEs with mid treatment termination	Cycle 1: grd 2 anemia; grd 3 supraventricular arrhythmias; grd 4 neutropenia. Cycle 2: grd 1 muscle weakness, neutropenia, and anemia. Cycle 3: grd 1 vomiting and anemia; grd 2 neuro-sensory. Cycle 4: grd 3 anemia and ataxia; grd 4 neuro-sensory. Off treatment cycle 4.	9	3	4	10	NA	6.5	26	4
**9. Severe** AEs with late treatment termination	Cycle 1: grd 1 myalgia; grd 2 hypoglycemia. Cycle 3: grd 1 anemia; grd 2 weight loss. Cycle 4: grd 1 constipation, anorexia, fatigue, stomatitis, and weight gain; grd 2 pain and anemia. Cycle 5: grd 1 constipation; grd 2 anemia and anorexia; grd 3 pain-bone and arrhythmia. Off treatment cycle 5.	3	0	3	9	11	5.2	26	3

**Table 2 cancers-12-03251-t002:** Comparison of overall AE burden score (*TB*) between active treatment and placebo.

**NCCTG 97-24-51**			
***TB***	**CAI** **(N = 85)**	**Placebo** **(N = 91)**	***p*-Value** **(Wilcoxon)**
Mean (SD)	6.1 (4.4)	3.9 (5.0)	
Median (range)	5.0 (0.0–24.0)	2.8 (0.0–37.0)	<0.0001
A091105			
***TB***	**Sorafenib** **(N = 49)**	**Placebo** **(N = 36)**	***p*-Value** **(Wilcoxon)**
Mean (SD)	4.4 (3.2)	3.1 (3.0)	
Median (range)	3.6 (0.7–18.5)	2.0 (0.0–13.5)	0.0042

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
