# Peer review of "Adverse Event Burden Score—A Versatile Summary Measure for Cancer Clinical Trials"

_cancers, 2020, doi:10.3390/cancers12113251_

Round 1
Reviewer 1 Report
Comments on ,”Adverse Event Burden Score – a Versatile Summary 3 Measure for Cancer Clinical Trials”
Authors defined indicators for each combination of k and g and a subjective weights to define burden score for cancer patients. Clearly, they tried to quantify and express pain and burden numerically, however the advantages of their suggested score over existing ones is not clear. The problem is that their score is subjective and can be manipulated by the investigators by varying the weights. Existing burden score like ESAS are standard and investigators can compare their samples and findings with others. This suggested adverse event burden score is not standardized and the findings of two different studies cannot be compared. Authors did not compare their burden score with the existing one. If there is any advantages or disadvantages using their burden score, it hasn’t been explored. The Appendix can be removed.
Author Response
Reviewer #1
Authors defined indicators for each combination of k and g and a subjective weights to define burden score for cancer patients. Clearly, they tried to quantify and express pain and burden numerically, however the advantages of their suggested score over existing ones is not clear. The problem is that their score is subjective and can be manipulated by the investigators by varying the weights. Existing burden score like ESAS are standard and investigators can compare their samples and findings with others. This suggested adverse event burden score is not standardized and the findings of two different studies cannot be compared. Authors did not compare their burden score with the existing one. If there is any advantages or disadvantages using their burden score, it hasn’t been explored. The Appendix can be removed.
Response: Thank you for your review. Our proposed method is flexible enough to accommodate a wide variety of different settings, and, in indeed, we view this flexibility as a strength of this methodology. However, we recommend that the weight function should be clearly defined a priori and emphasize that the same weight function should be used when comparing across trials or when comparing to an anchor. We have added clarifications (lines 118-123, 329-333) that the simple weight function used in our examples works in most settings and that more complex weight functions, if preferable, should be based upon a consensus from clinicians, patients, and experts.
As stated in the discussion, the advantages of the AE burden score include: i) it is a numerical summary which is more informative than the current standard of categorical summary of the maximum grade of the most common AEs (expounded in Subsection 2.2) and ii) AE burden score explicitly quantifies the overall AE severity, distills a sizable amount of data into a single score, and facilitates formal comparison between study arms. We agree that this approach needs to be clearly defined a priori to ensure its objectivity and its comparability across trials.
We believe that Appendix A is useful as the link between the proposed AE burden score and the current convention of reducing AE data to a categorical summary measure. As an appendix, it does not interfere with the flow of the main text and it provides further details for the interested readers. If the reviewer and editor feel strongly, we will remove Appendix A.
Reviewer 2 Report
Dear authors:
This is a very good manuscript with fine goals and application values.
The description of AE burden score in methods would be the most important part of this work. In addition to the statements of each parameter and equation, a figure or list including all parameters and equations is suggested to help the readers understand their relation and interaction more clearly.
Author Response
Reviewer #2
This is a very good manuscript with fine goals and application values.
The description of AE burden score in methods would be the most important part of this work. In addition to the statements of each parameter and equation, a figure or list including all parameters and equations is suggested to help the readers understand their relation and interaction more clearly.
Response: We are grateful for your positive comments and thank you for your review. We appreciate your suggestion. We have now added a list of parameters and equations to the end of the manuscript.
Round 2
Reviewer 1 Report
I have no further comments.